# Respiratory and Intestinal Microbiota in Pediatric Lung Diseases—Current Evidence of the Gut–Lung Axis

**DOI:** 10.3390/ijms23126791

**Published:** 2022-06-18

**Authors:** Sebastian Stricker, Torsten Hain, Cho-Ming Chao, Silvia Rudloff

**Affiliations:** 1Department of Pediatrics, Justus Liebig University Giessen, 35392 Giessen, Germany; silvia.rudloff@ernaehrung.uni-giessen.de; 2Institute of Medical Microbiology, Justus Liebig University Giessen, 35392 Giessen, Germany; torsten.hain@mikrobio.med.uni-giessen.de; 3German Center for Infection Research (DZIF), Partner Site Giessen-Marburg-Langen, Justus Liebig University Giessen, 35392 Giessen, Germany; 4Department of Pediatrics, University Medical Center Rostock, 18057 Rostock, Germany; cho-ming.chao@med.uni-rostock.de; 5Department of Nutritional Science, Justus Liebig University Giessen, 35392 Giessen, Germany

**Keywords:** gut–lung axis, respiratory microbiota, intestinal microbiota, short-chain fatty acids, toll-like receptors, COVID-19

## Abstract

The intestinal microbiota is known to influence local immune homeostasis in the gut and to shape the developing immune system towards elimination of pathogens and tolerance towards self-antigens. Even though the lung was considered sterile for a long time, recent evidence using next-generation sequencing techniques confirmed that the lower airways possess their own local microbiota. Since then, there has been growing evidence that the local respiratory and intestinal microbiota play a role in acute and chronic pediatric lung diseases. The concept of the so-called gut–lung axis describing the mutual influence of local microbiota on distal immune mechanisms was established. The mechanisms by which the intestinal microbiota modulates the systemic immune response include the production of short-chain fatty acids (SCFA) and signaling through pattern recognition receptors (PRR) and segmented filamentous bacteria. Those factors influence the secretion of pro- and anti-inflammatory cytokines by immune cells and further modulate differentiation and recruitment of T cells to the lung. This article does not only aim at reviewing recent mechanistic evidence from animal studies regarding the gut–lung axis, but also summarizes current knowledge from observational studies and human trials investigating the role of the respiratory and intestinal microbiota and their modulation by pre-, pro-, and synbiotics in pediatric lung diseases.

## 1. Introduction

The development of the human intestinal microbiota and its impact on health and disease is one of the most studied topics of the last decade. Every human mucosal surface is populated by a more or less complex and diverse microbiota that represents a key player in forming immune homeostasis. The fact that the count of bacterial cells (3.8 × 10^13^) outnumbers human cells (3.0 × 10^13^) within our body (1–10:1) further stresses the importance of the microbiome [1]. In animal models using germ-free mice, it was shown that a complete lack of commensal bacteria results in reduced viability due to impaired immune defense against pathogens and increased autoimmunity [2,3].

The individual composition of the microbiota is determined by endogenous (e.g., host genotype) and exogenous (mode of delivery, antibiotic exposure, diet) factors. Once established, the individual composition of the microbiota as well as its metabolic pathways remain stable [4]. The intestinal microbiota is the most studied part of the human microbiome, as it harbors the majority of bacteria and can be investigated by non-invasive sample collection using feces. Furthermore, the intestinal microbiota can be modified by several mechanisms including pre-, post-, and synbiotics. Prebiotics are non-digestive short chain carbohydrates that promote growth and/or activity of beneficial bacteria, whereas probiotics are defined according to the World Health Organization (WHO) as “live microorganisms which when administered in adequate amounts confer a health benefit on the host” [5]. The term "synbiotic" describes the combination with a prebiotic that specifically favors the accompanying probiotic.

Our knowledge about the composition of the microbiota was limited for a long time, as research was dependent on culture-based detection of microorganisms. However, next-generation sequencing techniques have been applied more and more in this field and enabled researchers to study the microbiome of organs (i.e., the lung) with minor bacterial colonization. It further led to the identification of many bacteria that cannot be cultivated, resulting in a much more detailed and complex image of the human microbiota.

There is growing evidence that the respiratory microbiota is altered in human lung diseases. Whether this is a cause or a consequence cannot always be clearly determined, but in some cases, lung dysbiosis was found to be associated with worse disease outcome. In addition, the intestinal microbiota plays a role in respiratory diseases, as intestinal dysbiosis in early infancy was shown to increase the risk of pulmonary infections and childhood asthma [6]. The inter-organ cross talk between the gut and the lung was named the “gut–lung axis”, and knowledge about the underlying mechanisms such as production of short chain fatty acids (SCFA) and signaling through Toll-like receptors (TLR) by the intestinal microbiota has grown recently [7].

In this article, we aim to summarize current mechanistic evidence from animal studies as well as observational data and clinical trials with reference to the role of the pulmonary and intestinal microbiota and their cross talk in pediatric lung diseases.

## 2. Composition and Development of Intestinal and Pulmonary Microbiota

### 2.1. Diversity and Development of Intestinal Microbiota

The human gut bacteria represent the vast majority of the microbiota, counting more than 10^13^ cells and encoding for more than three million genes, whereas the human genome solely comprises 23,000 genes. Current knowledge about intestinal microbiota predominantly focuses on fecal microbiota, as it is easy to acquire, exhibits a high microbial load (10^11^–10^12^ colony-forming units per ml content) and adequately reflects the composition of the colon microbiota. Contrarily, data about small bowel microbiota are sparse, as it can only be accessed by endoscopy and has a much lower microbial load (10^3^–10^4^ colony-forming units per ml content) [1]. The fecal microbiota of adult healthy humans is composed of about 150 different species, of which about 80 species account for a core microbiota that can be found in most individuals. The dominating phyla are *Firmicutes* (40%) and *Bacteroidetes* (20%), whereas *Proteobacteria*, *Actinobacteria* and others only represent a minority [4,8]. The intestinal microbiota is shaped within the first years of life and represents a vulnerable entity within this time period. Its composition is determined by numerous factors including the birth mode, breast-feeding practices, antibiotic exposure, and diet [6,9]. Prospective data suggest breast-feeding as the most important influence for gut bacteria composition, leading to increased relative abundance of *Bifidobacterium* species. Cessation of breast-feeding results in maturation of the microbiota, marked by elevated amounts of *Firmicutes* [10,11]. Another major influence on intestinal microbiota is the birth mode, as vaginal delivery is associated with increased levels of *Bacteroides* (in particular, *B. fragilis*) and faster maturation of the microbiota [11]. Current data indicate that the intestinal microbiota evolves within the first three years of life and remains stable after that with high interindividual variability [6,12]. Still, in addition to aforementioned exogenous factors, intestinal microbiota composition is also influenced by the host’s genotype. Twin studies revealed that monozygotic twins have a more similar microbiota than dizygotic twins [13]. Moreover, the LCT (lactase) gene haplotype associated with lactase non-persistence is associated with increased levels of *Bifidobacteria*, which are able to metabolize lactose [14]. Another association of the host genotype and the composition of the microbiota was reported for celiac disease. *Olivares* et al. showed that vaginally delivered and exclusively breastfed newborns carrying the HLADQ2 genotype that predisposes for celiac disease are colonized by a significantly different microbiota compared to infants not carrying this genotype. In detail, newborns with celiac disease predisposition displayed higher proportions of *Firmicutes* and *Proteobacteria* (e.g., enterotoxic *E. coli*) and corresponding lower population densities of *Actinobacteria* (especially *Bifidobacterium*) at 1 month [9,15]. These data indicate that genetic factors influencing metabolism and immune response of the host contribute to the variability in the composition of the intestinal microbiota.

It is known that gut bacteria not only contribute to intestinal metabolism and immune homeostasis, but also exert multiple systemic effects. The role of the intestinal microbiota was demonstrated in inflammatory bowel disease [16], allergy [17], autoimmunity [18], metabolism including obesity and type 2 diabetes [19], behavioral disorders [20] and cardiovascular diseases [21,22].

To conclude, the intestinal microbiota is a complex system that is influenced by multiple factors throughout its development within the first years of life with major implications for human health.

### 2.2. Diversity and Development of Respiratory Microbiota

Compared to the intestinal microbiota, data about the composition of bacterial species in the lung are sparse. The lower respiratory tract was considered sterile for a long time until next-generation sequencing techniques revealed bacterial genes in the lungs of healthy individuals. When looking at the respiratory microbiota, it is important to differentiate the sampling method, as the microbiota of the upper airways (e.g., nasopharyngeal swab, sputum) differs from the microbiota of the lower airways (e.g., bronchoalveolar lavage, lung tissue) [23]. The bacterial load (10^3^–10^5^ colony-forming units per mg lung tissue) and diversity of the respiratory microbiota are far less complex than that within the gut, but the most abundant phyla are the same, namely *Bacteroidetes* and *Firmicutes* [24,25,26,27]. On the genus level, *Pseudomonas*, *Streptococcus*, *Prevotella*, *Fusobacteria* and *Veillonella* dominate [28,29]. Compared to the intestinal microbiota, much less is known about the development of the composition of bacteria in the airways. There is some evidence, mostly deduced from murine models, that the respiratory microbiota maturates within childhood, and that this process is important in promoting tolerance to aeroallergens [30,31,32]. The bacterial composition within the lower airways is shaped by immigration (inhalation, microaspiration), elimination (cough, mucociliary clearance, host defense) and local growth conditions (oxygen, availability of nutrients such as carbon sources and iron) [23,25,33]. Alteration of these factors in the context of respiratory diseases contributes to pulmonary dysbiosis. In chronic inflammatory lung diseases such as cystic fibrosis (CF) or chronic obstructive pulmonary disease (COPD), an outgrowth of *Gammaproteobacteria* (e.g., *Pseudomonas aeruginosa*) was described. Factors promoting the expansion of *Pseudomonas aeruginosa* in chronic lung diseases include the production of reactive oxygen species during inflammation, reduced oxygen saturation and the fermentation of mucins by commensals, which generates propionate [33,34,35]. The composition of the respiratory microbiota is influenced by the birth mode and breast-feeding; however, the impact of both factors seems to be less than on the intestinal microbiota [36,37,38].

The role of respiratory microbiota composition was demonstrated in several lung diseases including asthma [39], bronchopulmonary dysplasia [40], cystic fibrosis [41] and respiratory infections including COVID-19 [42]. In some diseases, it was also shown that altered respiratory microbiota precedes disease onset, further stressing the pathophysiological role of the respiratory microbiota.

## 3. Mechanisms of Local and Systemic Immune Modulation by the Intestinal Microbiota

Even though the gut and the lung are two well-separated organs with their own local immune systems, there is much evidence that the intestinal microbiota is able to influence systemic immune homeostasis and specifically pulmonary immune regulation by several mechanisms that include (Figure 1):Production of short chain fatty acids (SCFA)Signaling through pattern recognition receptors (PRR)Segmented filamentous bacteria (SFB)

**Figure 1 ijms-23-06791-f001:**
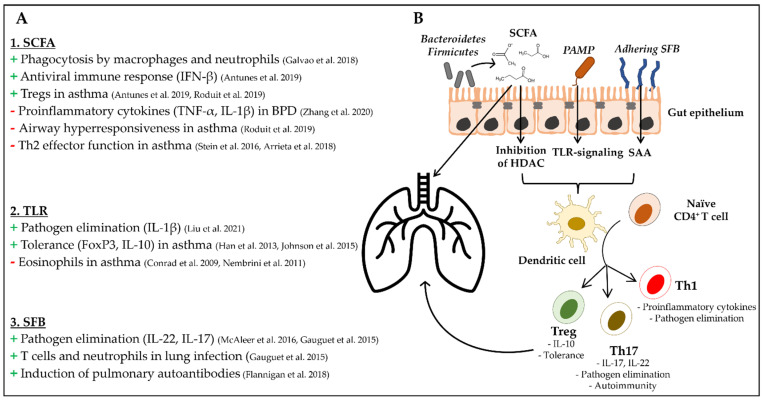
Major elements of cross talk between the gut and the lung. (**A**) Effects of SCFA, TLR signaling and SFB signaling on pulmonary immune mechanisms. (**B**) PAMP modulate dendritic cell signaling by TLR signaling, whereas SFB induce the production of SAA by epithelial cells, which modulates dendritic cell function. SCFA regulate dendritic cell signaling by direct inhibition histone deacetylases, which modulates gene expression. Dendritic cells migrate to intestinal lymph nodes and initiate differentiation of naïve CD4^+^ T cells and the acquisition of homing receptors [43,44,45,46,47,48,49,50,51,52,53,54,55,56,57].

Treg, Th17 and Th1 cells exert effector functions at distal sites including the lung. Systemic SCFA reach the lung via blood stream and influence pulmonary immune mechanisms by direct signaling via G protein-coupled receptors (GPR41/43). HDAC, histone deacetylases; SAA, serum amyloid A.

### 3.1. Production of Short Chain Fatty Acids (SCFA)

SCFA are saturated unbranched fatty acids including acetate, propionate and butyrate and are produced by anaerobic bacteria. The most prominent butyrate-producing bacteria belong to the phyla *Firmicutes* (e.g., *Faecalibacterium prausnitzii*, *Clostridia*), and *Bacteroidetes*. Moreover, *Bifidobacteria*, which are often used as probiotics, are able to produce acetate and lactate during carbohydrate fermentation. It is known that SCFA exert multiple beneficial effects within the intestinal mucosa. They (especially butyrate) serve as the primary energy source for colonocytes and exert diverse local anti-inflammatory effects by increasing mucus production by goblet cells [58,59], enhancing intestinal barrier integrity [60,61,62], inducing tolerogenic dendritic cells and regulatory T cells [43,63,64], and increasing sIgA production [65,66,67] and production of antimicrobial peptides [68,69].

In addition to those local effects, SCFA (especially acetate) reach the blood circulation and peripheral tissues, and even pass the blood–brain barrier [19]. Their signaling at distal organ sites is mediated by G protein-coupled receptors including GPR41, GPR43 and GPR109. Further, butyrate and propionate are known to directly inhibit histone deacetylases, which modifies gene expression, resulting in multiple downstream effects [70]. Their systemic effects on human metabolism have been demonstrated within the context of the so-called gut–brain axis. In mice, oral butyrate substitution was demonstrated to reduce appetite and energy intake and to increase fat oxidation, which results in the prevention of diet-induced obesity [71]. Data from animal and human trials indicate that SCFA induce the production of anorectogenic glucagon-like peptide 1 and peptide yy by signaling through GPR43, thus exerting beneficial effects on host metabolism [72,73].

Gut bacteria-derived SCFA were further shown to exert multiple effects on the immune system including the lung [44]. Butyrate and propionate induce peripheral forkhead box protein P3 (FoxP3+) regulatory T cells by direct interaction with CD4^+^ T cells and dendritic cells via inhibition of histone deacetylases [43]. Butyrate further reduces the expression of co-stimulatory surface molecules and impairs T cell activation by dendritic cells [64,66]. SCFA also play a role in neutrophil migration. In rodents, expression of the adhesion molecule L-selectin and migration of neutrophils to the subcutaneous tissue was stimulated by SCFA (acetate < butyrate < propionate) under non-inflammatory conditions, whereas recruitment of neutrophils to the peritoneum and expression of TNF-α was reduced by SCFA (butyrate and propionate) [70,74,75]. In addition, SCFA contribute to pathogen elimination, as they induce the expression of antimicrobial peptides by epithelial cells (Figure 1A,B). In rodents, enteral administration of acetate and butyrate has been shown to induce the expression of β-defensins and cathelicidin-related antimicrobial peptide in a GPR43-mediated manner in intestinal epithelial cells [69]. Treatment with butyrate also stimulated the production of cathelicidin by human lung epithelial cells [76]. GPR43-mediated signaling of acetate was further shown to be essential for adequate phagocytosis of pathogens by alveolar macrophages and neutrophils during *Klebsiella pneumoniae* infection and for IFN-β-mediated antiviral immune response during respiratory syncytial virus infection in mice [45,46] (Figure 1A,B).

In conclusion, SCFA represent a key element of the interaction between commensals and the host and exert pleiotropic beneficial effects on metabolism and the immune system.

### 3.2. Signaling through Pattern Recognition Receptors (PRR)

The intestinal microbiota is in a close functional relationship with intestinal immune cells. So-called pathogen-associated molecular patterns (PAMP), which are highly conserved microbial components, are recognized by specialized PRR of innate immune cells. The best known PRR are transmembraneous Toll-like receptors (TLR) that sense specific bacterial components (e.g., TLR2—peptidoglycans, TLR4—lipopolysaccharide (LPS), TLR5—flagellin/profilin, TLR 9—bacterial DNA) and initialize signal cascades that contribute to intestinal immune homeostasis [77,78]. TLR signaling is a part of the complex interaction between commensal bacteria within the intestinal microbiota and the host [79]. Downstream effects of TLR signaling can promote tolerance (e.g., by enhanced expression of IL-10 or transdifferentiation to regulatory T cells) as well as immune response against pathogens (e.g., activation of NF-κB or expression of IL-17, IL-22; Figure 1A,B) [80]. The type of immune response (pro- or anti-inflammatory) depends on many factors, including dimerization with co-receptors, expression of negative regulators (toll-interacting protein, TOLLIP) [81], localization of the TLR (apical vs. basolateral) and concentration of the ligand [82]. The local importance of TLR signaling for immune homeostasis within the gut mucosa has been extensively demonstrated in murine models [83,84]. Modification of systemic immune mechanisms is achieved by dendritic cells that recognize PAMP, migrate to the mesenteric lymph nodes and prime T cells, including the acquisition of homing receptors (e.g., CC chemokine receptor type 4, CCR4), which leads to the migration to different body sites including the respiratory mucosa [85,86] (Figure 1B). In the context of the gut–lung axis, it was shown that gut bacteria-derived LPS, a potent PAMP, modulates inflammatory response by inducing expression of IL-1β in alveolar macrophages and alveolar type 2 cells [44] (Figure 1A). The role of TLR signaling becomes more clear regarding the fact that TLR-deficient mice exhibit a reduced microbial biomass and diversity as well as an altered composition of the respiratory microbiota compared to wild-type mice [87]. LPS-mediated TLR4 signaling is essential in host defense against bacterial pneumonia, as demonstrated in murine models [88,89]. Further, TLR signaling also plays a role in inducing tolerance in allergic airway disease in rodents [47,48]. Mice with defective TLR4 signaling show reduced ability to induce FoxP3 expression by T cells and induction of regulatory T cells, all mechanisms that are associated with immune tolerance [49].

### 3.3. Segmented Filamentous Bacteria (SFB)

Segmented filamentous bacteria (SFB) represent specific commensal bacteria related to the genus *Clostridium* that attach closely to the epithelium of the terminal ileum [50]. SFB penetrate the intestinal mucus layer and directly interact with epithelial and immune cells without invading the host (Figure 1B). They were shown to induce the production of sIgA [90,91] and to play a major role in the maturation of the intestinal immune system in terms of differentiation of T cells (Th1, Th2, Th17 and regulatory T cells) [92]. One specific subset of T cells that was associated with SFB is antigen-specific, pro-inflammatory Th17 CD4^+^ T cells, which are characterized by IL-17 production, a cytokine that is known to directly activate enterocytes, leading to the secretion of antimicrobial peptides and the recruitment of neutrophil granulocytes. The induction of Th17 cells by SFB thus prevents intestinal colonization by pathogens. However, the effects of SFB on the immune system of the host appear to exceed local effects. In mice, SFB colonization was shown to prevent expansion of methicillin-resistant *Staphylococcus aureus* (MRSA) and *Aspergillus fumigatus* within the lung by Th17 cells and IL-22-mediated mechanisms [51,52]. On the other hand, there are some data indicating that pulmonary viral infection reduces the number of SFB in the intestinal microbiota [93] (Figure 1A). However, stimulation of the immune system by SFB does not always result in beneficial effects. In mouse models, it was shown that colonization of germ-free mice with SFB triggers rheumatoid arthritis as well as multiple sclerosis in a Th17-mediated manner [50]. Further data indicated that in mice that are prone to autoimmunity, Th17 cells that were induced by SFB are recruited to the lung by the chemoattractant CCL20 and induce autoimmune lung pathology [94].

These observations further stress the fact that the effects of the intestinal microbiota can vary depending on the immune status of the host. The interactions between gut bacteria and the human immune system are complex, with multiple influencing variables.

## 4. Respiratory Microbiota and Gut–Lung Axis in Human Diseases

### 4.1. Bronchopulmonary Dysplasia (BPD)

BPD is a chronic lung disease caused by pre- and postnatal injury of the developing lung of preterm infants and represents one of the major morbidities after preterm birth [95]. Several circumstances associated with prematurity lead to a disrupted development of the microbiota. Preterm neonates are often delivered by cesarean section, often need extensive antibiotic treatment, and suffer from reduced enteral food intake and reduced skin contact. The role of the intestinal microbiota in necrotizing enterocolitis, another major complication of preterm birth, has been investigated throughout the last decades, and several measures promoting a beneficial intestinal microbiota such as enteral feeding of human milk or supplementation with probiotics have already reached clinical practice [96,97].

The evidence for the effect of the host microbiota on BPD is much less explored to date (Table 1). The most abundant bacteria in endotracheal aspirates of preterm infants are *Staphylococcus* spp. and *Ureaplasma* spp. [40]. Recent studies described a reduced prevalence of anti-inflammatory phyla such as *Lactobacilli* and increased abundance of proinflammatory phyla such as *Proteobacteria* and bacteria with pathogenic potential in preterm infants with BPD [98,99,100]. The role of *Ureaplasma urealyticum*, a microorganism that is a major cause of chorioamnionitis, in BPD remains controversial to date [101]. Experimental data indicate that colonization with this species might exert beneficial immunological effects, whereas single-center studies report inconsistent results concerning the risk of BPD in colonized preterm infants [102,103,104]. Antibiotic treatment of preterms colonized with *Ureaplasma urealyticum* with azithromycin reduced the risk of developing BPD; however, it is not clear whether this effect is due to the eradication of *Ureaplasma* or other effects (e.g., anti-inflammatory properties) of the macrolide therapy [105].

Concerning the role of the intestinal microbiota in BPD, current data indicate an association of increased abundance of the phyla *Proteobacteria,* and specifically of certain *Gammaproteobacteria* (*Escherichia/Shigella*, *Klebsiella and Salmonella*), with BPD in preterm neonates [121,122] (Table 2). Further, experimental evidence indicates a beneficial role of oral acetate substitution in reducing remodeling and tissue inflammation in a murine model of BPD [53]. Consistent with this, antibiotic exposure led to an increased proportion of *Proteobacteria* within the intestinal microbiota, which was associated with reduced survival rates and increased pulmonary inflammation in a BPD mouse model [123].

### 4.2. Cystic Fibrosis (CF)

Cystic fibrosis (CF) represents an autosomal recessive multi-systemic disease caused by mutations of the cystic fibrosis transmembrane conductance regulator (CFTR) gene. This leads to a dysfunctional epithelial anion transporter, resulting in thickened mucus in the pulmonary and gastrointestinal system.

One major complication of CF is the pulmonary colonization with specific pathogens such as *Pseudomonas aeruginosa*, *Staphylococcus aureus*, *Burkholderia cepacia complex* and *Stenotrophomonas maltophilia*, which contribute to progressive respiratory failure and display a major factor of morbidity of the disease [137]. In recent years, next-generation sequencing techniques have significantly increased the knowledge about the respiratory microbiota in CF. Several studies demonstrated an association of reduced respiratory microbiota diversity with impaired lung function [108,109] (Table 1). Furthermore, microbiota composition changes throughout the disease course. In younger children, CF respiratory microbiota already displays reduced diversity compared to healthy control patients. Their respiratory microbiota is dominated by *Streptococcus* and *Prevotella* [109]. Throughout the disease course, bacterial diversity decreases, and the respiratory microbiota tends to be dominated by typical CF pathogens such as *Pseudomonas aeruginosa* and others [110] (Table 1). Although, it is assumed that pulmonary dysbiosis may precede acute disease exacerbation, a distinct pattern of the composition of the respiratory microbiota has not been found yet [137]. Thus, it remains unclear to what extent pulmonary dysbiosis is caused by the CF genotype.

Since CF is a multi-systemic disease and often causes pancreatic insufficiency, it seems reasonable to assume that the composition of the intestinal microbiota is altered as well. Diversity and density of intestinal microbiota is significantly decreased in CF patients regardless of their pancreatic functionality [124,125,138] (Table 2). A longitudinal study compared the development of pulmonary and intestinal microbiota in infants with CF and reported that even though the lung and the gut were dominated by different genera, they shared a core microbiota consisting of *Veillonella* and *Streptococcus* [126,139]. The intestinal microbiota of CF patients is further characterized by an increased abundance of *Proteobacteria* and specifically *Escherichia coli/Shigella*, whereas the abundance of *Bacteroides*, *Ruminococcaceae* and *Lachnospiraceae* is reduced [124,127,128] (Table 2). The influence of the intestinal microbiota on the composition of the respiratory microbiota is further stressed by the fact that changes in the composition of the intestinal microbiota often preceded alterations of the respiratory microbiota [139]. The gut dysbiosis observed in CF has clinical importance, as it is associated with disease exacerbation, growth retardation and impaired lung function [125,126,127,128] (Table 2). Interestingly, it was shown that changes in the intestinal microbiota (i.e., reduction in *Parabacteroides*) but not in the respiratory microbiota preceded pulmonary colonization with *Pseudomonas aeruginosa* and CF exacerbation [126].

These observations indicate that modulating the intestinal microbiota might improve disease outcome of CF. In fact, breast-feeding of CF infants revealed a trend towards prolonged time to first pulmonary exacerbation [126]. Further, randomized trials indicate that a probiotic therapy can not only reduce fecal calprotectin as an intestinal inflammation parameter, but also decrease the frequency of pulmonary exacerbations in CF [140,141].

Even though human studies hint the therapeutic potential of modifying intestinal microbiota in CF, it is not clear whether this can improve the disease outcome. However, healthcare givers should be aware of this therapeutic option, especially in patients with markers of intestinal inflammation (e.g., fecal calprotectin) [138].

### 4.3. Respiratory Infectious Diseases

Host microbiota plays an important role in resistance to pulmonary pathogens. As shown in several murine models, germ-free mice lacking any bacteria are prone to acute bacterial and viral pulmonary infections [42]. Their immune response and survival can be improved by administration of microbial metabolites (SCFA), microbial components (PAMP, e.g., LPS) or probiotics. However, in most cases, those substances were given by the oral route (in part simultaneously with intranasal gavage), and alterations of the microbiota were observed in the gut as well as in the lung [142]. Hence, mechanistic data about the influence of respiratory microbiota on lung diseases are still sparse. In humans, alteration of respiratory microbiota was described in patients suffering from acute respiratory distress syndrome (ARDS). The prevalence of gut-associated bacteria such as *Enterobacteriaceae* and *Bacteroidetes* was significantly elevated in ARDS and sepsis patients and correlated with markers of lung inflammation and clinical outcome [112,113] (Table 1). A common hypothesis for the expansion of typical intestinal bacteria in the pulmonary environment is the phenomenon of the “leaky gut”, where increased permeability of the intestinal epithelial barrier in acute severe diseases facilitates translocation of intestinal bacteria. However, one has to keep in mind that pulmonary dysbiosis in critically ill patients might be as well caused by increased incidence of microaspirations, altered local microenvironment such as an increased mucus production or altered oxygen concentration in the lung and the treatment in the context of an intensive care unit (e.g., with antibiotics or proton pump inhibitors, parenteral feeding, etc.) [143].

The essential role of the intestinal microbiota in acute viral and bacterial pulmonary infections has been demonstrated in several murine models after reduction of the gut bacteria by extensive antibiotic treatment. It was shown that restoration of intestinal SFB improved disease outcome in pneumonia caused by *Staphylococcus aureus* [52]. In addition, several studies reported beneficial effects of the administration of *Bifidobacterium longum* on the immune response in murine Influenza A infection [144,145]. More mechanistically, TLR signaling and SCFA were shown to improve disease outcome and lung inflammation in different murine models [146,147]. One study further demonstrated that oral but not pulmonary administration of PRR was able to improve host defense against *Klebsiella pneumoniae*, which stresses the role of the gut–lung axis in infectious diseases [148]. Finally, it was also shown that pulmonary infections with influenza virus led to intestinal dysbiosis in mice that resulted in reduced levels of acetate. These reduced SCFA levels were associated with an increased susceptibility to secondary pulmonary infection [149].

The potential therapeutic role of modulating the intestinal microbiota in order to prevent and/or treat pulmonary infections was analyzed by King et al., evaluating twenty randomized controlled trials involving the administration of *Bifidobacterium* and *Lactobacillus* strains in humans. They reported a significantly reduced duration of acute respiratory infections in the treatment group (not always double-blind) in otherwise healthy participants [150]. In a randomized placebo-controlled trial, Luoto et al. showed that prebiotic (galacto-oligosaccharide and polydextrose mixture) or probiotic (*Lactobacillus rhamnosus LGG*) treatment reduces the incidence of upper respiratory tract infections in preterm neonates [151]. This observation was confirmed by Maldonado et al., who reported reduced incidence of respiratory and gastrointestinal infections in neonates treated with synbiotics (*Lactobacillus fermentum* and galacto-oligosaccharide) [152]. The only available meta-analysis (23 trials, 6269 children) on this topic, by Wang et al., states that probiotic supplementation reduces the incidence of upper respiratory tract infections in children with moderate quality of evidence [153]. A recent Cochrane review further demonstrated that probiotics are superior to placebo in reducing the incidence and duration of upper respiratory tract infections in adults [154]. In addition, probiotic supplementation may have a beneficial impact on the incidence of ventilator-associated pneumonia in critically ill patients. Several meta-analyses including a Cochrane review were conducted and consistently reported that the incidence of pneumonia was reduced; however, it remains unclear whether this effect goes along with reduced mortality and hospital stay [155,156,157,158].

### 4.4. SARS-CoV-2 and COVID-19

Severe acute respiratory syndrome coronavirus 2 (SARS-CoV-2) is known to affect not only the lung but also the gut, and fecal excretion of SARS-CoV-2 specific nucleic acids is often apparent in affected individuals. Gastrointestinal symptoms including loss of appetite, nausea and vomiting are frequent in Coronavirus Disease 19 (COVID-19) patients (20–60%), can precede pulmonary symptoms and may be associated with severe progress of the disease [159]. Experimental data have demonstrated the expression of angiotensin converting enzyme 2 (ACE2), the binding site of the SARS-CoV-2 spike protein, by human intestinal epithelial cells and the reproduction of the virus within the gastrointestinal tract [160,161].

Concerning respiratory microbiota, Heba et al. observed a decreased microbial diversity in SARS-CoV-2 positive individuals [114]. Further, respiratory microbiota from COVID-19 patients showed higher abundance of *Pseudomonas*, *Enterobacteriaceae* and *Acinetobacter*, whereas patients with non-COVID pneumonia had higher amounts of lung commensal bacteria [115] (Table 1). However, to date there are no data that clearly indicate an association of a distinct respiratory microbiota with disease outcome in COVID-19 [114,162].

With regard to gut bacteria, Yeoh et al. have shown that the composition of the intestinal microbiota not only differed between COVID-19 patients and controls, but was also associated with disease severity and markers of systemic inflammation [129]. In particular, *Faecalibacterium prausnitzii* and *Bifidobacterium bifidum*, both microorganisms with beneficial immunomodulatory properties, were reduced in COVID-19 patients and inversely correlated with disease severity [130] (Table 2). This observation was confirmed by Gu et al., who described a reduced bacterial diversity and a different composition of the intestinal microbiota in COVID-19 compared to influenza patients [131]. Another observation underlining the role of the intestinal microbiota in COVID-19 is the fact that the abundance of the phylum *Bacteroidetes*, known to downregulate ACE2 expression in the murine gut, was negatively correlated with the SARS-CoV-2 fecal load [130]. The intestinal dysbiosis seen in COVID-19 was associated with the presence of fecal SARS-CoV-2 and persisted even after the clearance of SARS-CoV-2 in the lung and the end of symptoms [129,130,160,163,164] (Table 2). Still, those observations have to be interpreted with caution, as most studies only compare COVID-19 patients with healthy controls or other control groups (e.g., community-acquired pneumonia) with significantly different clinical course.

Research and evidence on the potential benefits of probiotic treatment to prevent or treat COVID-19 are growing, but to date still limited. Retrospective single-center studies found reduced mortality, inflammatory biomarkers and duration of hospital stay in patients receiving additional probiotic medication [165,166,167]. However, a recent randomized open label trial using a multi-strain probiotic did not show any effects on mortality, morbidity or inflammatory biomarkers [168]. Another approach could be to prevent respiratory tract infection by oropharyngeal administration of probiotics, which has been shown to be efficient in reducing infections in a prospective open label trial [169]. It can be expected that the knowledge about pre- and probiotic therapeutic possibilities in COVID-19 will increase within the next years and may potentially reveal new drug candidates for the prevention or treatment of the disease.

### 4.5. Allergic Asthma and Atopy

Allergic asthma is a chronic inflammatory lung disease characterized by wheezing, acute broncho-obstruction and respiratory distress. T helper 2 (Th2) cells play a major role in asthma pathogenesis by secreting proinflammatory cytokines (IL-4/5/9/13), leading to the production of immunoglobulin (Ig) E, goblet cell differentiation and the activation of eosinophils. The resulting chronic inflammation causes broncho-pulmonary hyper-responsiveness and airway remodeling [170].

The role of the intestinal microbiota in shaping human immune response and preventing allergic diseases including asthma has been extensively investigated in the past, but recent evidence was also reported on a role of the respiratory microbiota.

Bisgaard et al. demonstrated that colonization of the upper airways with pathogens such as *Streptococcus pneumoniae*, *Moraxella catarrhalis* and *Haemophilus influenzae* at 1 month of age was associated with asthma at 5 years [116] (Table 1). Further analyses using next-generation sequencing revealed a positive association of the relative abundance of *Prevotella* and *Veillonella* with distinct chemokines and asthma prevalence [117]. In adult asthma patients, higher abundance of *Proteobacteria* in the lower respiratory tract was repetitively reported [118,119] (Table 1). Further, the presence of *Proteobacteria* negatively correlated with pulmonary eosinophil count but was associated with increased expression of proinflammatory Th17-related genes [120]. Those observations indicate that respiratory microbiota might play a role in asthma as well. However, it has to be considered that the observed lung dysbiosis of adult asthma patients might rather be a consequence of lung inflammation or treatment with inhalative corticosteroids than a causal factor [171,172]. In addition, the aforementioned studies did not investigate the composition of the intestinal microbiota; thus, it remains open whether the observed differences are specific for the lung.

The role of the intestinal microbiota in atopy has gained much attention in recent decades as part of the hygiene hypothesis, which states that a lack of early life exposure to microbial components impairs the maturation of the human immune system and increases susceptibility to atopic diseases [85]. Observational data indicate that intestinal dysbiosis in infancy, especially during the so-called critical “window of opportunity”, is associated with atopy and asthma [85]. The fact that reduced diversity of the intestinal microbiota as well as antibiotic treatment within the first year of life were associated with increased risk of asthma further underlines the importance of the microbiota in early life [133,134,135,173]. A protective role against atopic diseases is also attributed to bacterial components within animal sheds and unprocessed cow’s milk, which is described by the so-called “farm effect”. It seems probable that microbial components stimulate and shape innate immune response towards a tolerogenic phenotype. However, currently it is not known which specific components confer this effect. Even though it was shown that a farm environment influences maturation of intestinal microbiota, it is still not fully elucidated whether inhalation or ingestion is the main route for immunomodulatory bacterial components [54,174,175,176,177,178].

The role of the intestinal microbiota in atopy is further stressed by the fact that an increased prevalence of *Streptococcus* species and a reduced abundance of *Bifidobacterium* and *Faecalibacterium* in the feces of young infants were associated with asthma prevalence at school age [7,55,136] (Table 2). Those observations could be explained by beneficial effects mediated by SCFA. It has been demonstrated that reduced amounts of fecal acetate and butyrate in the first year of life were associated with asthma at school age, whereas high levels of fecal butyrate and propionate correlated with reduced prevalence of atopy in later life [7,55,176] (Table 2). Using a murine model, Roduit et al. demonstrated that oral administration of SCFA reduced methacholine-induced airway hyperresponsiveness and increased the number of regulatory T cells in bronchoalveolar lavage fluid [56] (Figure 1A). In various other models of murine allergic airway inflammation, SCFA were shown to exert protective immunomodulatory effects by reducing proinflammatory Th2 immune response, promoting tolerogenic functions of bone marrow-derived dendritic cells, and inducing regulatory T cells [179,180] (Figure 1A,B). Thorburn et al. further demonstrated that a high-fiber diet leads to beneficial effects associated with increased acetate levels in a murine asthma model [181]. Finally, acetate supplementation or high-fiber diets in pregnant mice reduced indicators of allergic airway disease in the adult offspring. The fact, that this was not observed when pregnant mice were treated with SCFA or high-fiber diet after giving birth and throughout lactation indicates that these effects were mediated in utero [181]. Another potential player in asthma might be polysaccharide A (PSA) (a strong TLR-2 agonist) from *Bacteroides fragilis*, which has been associated with immunoregulatory functions in several diseases. Johnson et al. demonstrated that oral administration of PSA reduced the susceptibility to allergic asthma by CD4^+^ T cells and increased levels of the anti-inflammatory cytokine IL-10 [57].

In human trials, it was shown that the administration of *Bifidobacterium infantis* over 8 weeks increased peripheral IL-10 concentrations and FoxP3^+^ T cells, a mechanism that was shown to be mediated by dendritic cells in vitro [182] (Figure 1A). Treatment with synbiotics reduced biomarkers of Th2 inflammation and was associated with improved lung function in asthmatic children. However, it remains unclear which specific part of the intervention (pre- or probiotic) caused the observed effects [183,184]. A single synbiotic treatment with inulin and *Lactobacilli* reduced exhaled nitric oxide concentration and inflammatory biomarkers in induced sputum in adult asthma patients [185]. Several randomized trials have been conducted to investigate the effect of syn- and probiotic treatment in early life on allergic sensitization and asthma. Even though the risk of atopic dermatitis and allergic sensitization was reduced in later life, the risk of developing asthma was not affected [186,187,188]. Moreover, in several meta-analyses, no sufficient evidence for a preventive effect of probiotics was found, but there was at least some evidence that pre- and synbiotic treatment can reduce the risk of developing asthma and that probiotics can exert beneficial effects on inflammation and pulmonary function in asthma patients [189,190,191,192,193].

To conclude, current data indicate that pre- and probiotic therapy are able to reduce allergy-associated biomarkers and atopic sensitization, but the beneficial effects on definite asthma remain unclear [192,194]. Thus, current guidelines from scientific societies do not recommend pre-, pro-, or synbiotic therapy in order to prevent allergy due to the lack of scientific evidence [195,196]. Future research will clearly aim at identifying the optimal time point and the optimal instrument for modulating the microbiota in the context of allergic disease.

## 5. Conclusions

Recent advances in the understanding of host–microbe interactions, especially within the intestine, have highlighted the importance of intestinal microbiota and the gut–lung axis in pediatric lung diseases. Even though knowledge is growing, little is known about the microbiota of the lower airways. Novel sequencing methods revealed bacterial colonization of the lung in healthy individuals and paved the way for the investigation of the respiratory microbiota in human diseases. Even though current data indicate that the airway microbiota is altered in some respiratory diseases such as CF, BPD and asthma, it is not completely clear whether the observed differences are a cause or a consequence. Future research should aim at clarifying the pathophysiological role of the respiratory microbiota, as modulation of lung bacteria might be a therapeutic tool in some diseases.

In contrast to the respiratory microbiota, much more is known about the intestinal microbiota and its interaction with the human immune system. The intestinal microbiota shapes lung immunity not only by producing SCFA derived from fermentation processes but also influences systemic immune response by intestinal TLR signaling and SFB-mediated priming of T cells. However, the fact that commensal SFB do not only enhance pathogen elimination but also play a role in Th17-mediated autoimmunity shows that the interaction between the immune system and the microbiota is a complex mechanism. Currently, there is not much evidence from human trials and meta-analyses that clearly demonstrates the benefit of microbiota modulation by pre-, pro-, or synbiotics in respiratory diseases. Still, it can be expected that the application of next-generation sequencing techniques will substantially improve our knowledge about gut–lung interaction. Future research should aim at further clarifying the potential of microbiota modulation in the prevention and treatment of lung diseases.

## Figures and Tables

**Table 1 ijms-23-06791-t001:** Respiratory microbiota in pediatric lung diseases. Age statements are shown either as mean ± standard deviation or as median and range.

Lung Disease	Study Cohort	Sampling	Sequencing	Results	Ref
BPD	67 preterm neonates with no or mild BPD, age 27 (24–31) gestational weeks, 35 preterm neonates with severe BPD, age 26 (24–29) gestational weeks	Nasopharyngeal swabs	Cultivation, MALDI-TOF	- Higher frequency of pathogenic bacteria in infants with severe BPD	[99]
	10 preterm neonates with BPD, age 26 ± 2 gestational weeks, 12 preterm neonates without BPD, age 29 ± 1 gestational weeks	Endotracheal aspirates	16S rRNA sequencing (V3/V5 region)	- Reduced bacterial diversity in patients developing BPD	[100]
	55 preterm neonates, age 26 (23–30) gestational weeks	Endotracheal aspirates	Species-specific PCR	- Higher risk of BPD or death in the presence of *Ureaplasma*	[104]
	155 preterm neonates, 51 with mild BPD, age 26 ± 1 gestational weeks, 49 with moderate BPD, age 26 ± 2 gestational weeks, 52 with severe BPD, age 25 ± 2 gestational weeks	Endotracheal aspirates	16S rRNA sequencing (V1/V2 region)	- Higher abundance of *Ureaplasma* and lower abundance of *Staphylococcus* in severe BPD	[106]
	23 preterm neonates, age 25 ± 0 gestational weeks, 10 full term neonates, age 38 ± 2 gestational weeks	Endotracheal aspirates	16S rRNA sequencing (V4 region)	- Reduced abundance of *Firmicutes* and genus *Lactobacillus* and increased abundance of *Proteobacteria* in neonates developing BPD	[107]
CF	57 CF patients with moderate lung disease, age 31 ± 10 years, 139 CF patients with moderate lung disease, age 30 ± 10 years, 101 CF patients with severe lung disease, age 30 ± 11 years	Sputum	16S rRNA sequencing (V1–V2 region)	- Association of reduced bacterial diversity and increased abundance of CF pathogens with impaired lung function in CF patients	[108]
	76 pediatric CF patients, age 12 (4–17) years, 193 adult CF patients, age 29 (18–64) years	Sputum	Si-Seq 16S rRNA sequencing	- Core respiratory microbiota consisting of *Streptococcus*, *Prevotella*, *Rothia*, *Veillonella* and *Actinomyces* - Higher bacterial diversity in younger patients - Association of low bacterial diversity with impaired lung function - Domination of lung microbiome by CF pathogens	[109]
	51 control patients, age 6 ± 4 years, 101 CF patients, age 4 ± 2 years	Bronchoalveolar lavage	16S rRNA sequencing (V3/V4 region)	- Non-CF respiratory microbiota diversity increases with age - CF respiratory microbiota diversity decreases with age - Abundance of *Staphylococcus* and *Pseudomonas* increases with age in CF patients - Abundance of *Streptococcus*, *Veillonella* and *Porphyromonas* decreases with age in CF patients	[110]
	21 infants with CF, age 2 (2–3) months, 10 infants without CF, age 6 (3–9) months	Bronchoalveolar lavage	16S rRNA sequencing (V1–V3 region)	- Bacterial biomass was associated with inflammation - Reduced bacterial diversity in CF patients - Increased abundance of *Staphylococcus* and reduced prevalence of *Fusobacterium* in CF patients	[111]
Respiratory infectious diseases and COVID-19	91 critically ill patients, age 61 ± 16 years	Bronchoalveolar lavage	16S rRNA sequencing	- Longer ventilation of patients with increased bacterial burden - Association of the abundance of gut-specific bacteria (*Lachnospiraceae* and *Enterobacteriaceae*) with duration of ventilation and incidence of ARDS	[112]
	68 patients with ARDS, age 47 ± 15 years	Bronchoalveolar lavage	16S rRNA sequencing (V4 region)	- Enrichment of respiratory microbiota with gut-specific bacteria (*Bacteroidetes*) was associated with severity of systemic inflammation	[113]
	40 COVID-19 positive patients, 10 COVID-19 negative patients, age 51 (18–78) years	Nasopharyngeal swab	Metagenomic sequencing	- Reduced microbial diversity in COVID-19 patients - Higher abundance of *Propionibacteriaceae* and reduced abundance of *Corynebacterium accolens* in COVID-19-negative patients	[114]
	24 patients with COVID-19 pneumonia, age 68 (59–71) years, 24 patients with non-COVID-19 pneumonia, age 64 (50–71) years	Bronchoalveolar lavage	16S rRNA sequencing (V3/V4 region)	- Lower abundance of lung commensal bacteria in COVID-19 patients	[115]
Asthma	321 neonates at risk of developing asthma, age 1 month	Hypopharyngeal aspirate	cultivation	- Increased risk of wheeze and asthma, elevated eosinophil counts and total IgE in later life of infants colonized with *H. influenzae*, *S. pneumoniae*, *M. catarrhalis*	[116]
	695 infants at risk of developing asthma, sampling at age of 1 week, 1 month and 3 months	Hypopharyngeal aspirate	16S rRNA sequencing (V4 region)	- Association of asthma at the age of 6 years with higher relative abundance of *Prevotella* and *Veillonella.*	[117]
	10 asthma patients, age 26 ± 1 years, 10 non-asthmatic patients, age 26 ± 1 years	Induced Sputum	16S rRNA sequencing (V6 region)	- Higher relative abundance of *Proteobacteria* in asthma patients	[118]
	12 healthy patients, age 35 ± 10 years, 18 patients with non-severe asthma, age 45 ± 16 years, 26 patients with severe asthma, age 48 ± 11 years	Induced Sputum	16S rRNA sequencing (V3/V5 region)	- Reduced relative abundance of *Bacteroidetes* and *Fusobacterium* and increased relative abundance of *Proteobacteria* (non-severe asthma) and *Firmicutes* (severe asthma) in asthmatic patients	[119]
	40 patients with severe asthma, age 46 (20–63) years	Bronchial brushing	16S rRNA sequencing	- Higher relative abundance of *Actinobacteria* in patients with severe asthma compared to controls and patients with mild to moderate asthma (previously studied cohorts)	[120]

**Table 2 ijms-23-06791-t002:** Intestinal microbiota in pediatric lung diseases. Age statements are shown either as mean ± standard deviation or as median and range.

Lung Disease	Study Cohort	Sequencing	Results	Ref
BPD	20 preterm neonates with BPD, age 26 ± 2 gestational weeks, 28 preterm neonates without BPD, age 27 ± 1 gestational weeks	16S rRNA sequencing (V4 region)	- Increased relative abundance of *Escherichia/Shigella* and reduced relative abundance of *Klebsiella* and *Salmonella* in vaginally delivered preterm neonates with BPD	[121]
	18 preterm neonates with BPD, age 27 ± 2 gestational weeks, 20 preterm neonates without BPD, age 30 ± 1 gestational weeks	16S rRNA sequencing (V4 region)	- Reduced bacterial diversity in neonates with BPD - Increased relative abundance of *Proteobacteria* and reduced relative abundance of *Firmicutes* in neonates with BPD	[122]
CF	35 healthy control patients, 3 CF patients with pancreatic sufficiency, 20 CF patients with pancreatic insufficiency, age 1–17 years	16S rRNA sequencing (V1–V3 region)	- Reduced richness and diversity of intestinal microbiota in CF patients independent of pancreatic function - Reduced abundance of *Ruminococcaceae* and *Lachnospiraceae* in CF patients - Increased abundance of *Escharichia/Shigella* in CF patients	[124]
	409 healthy infants, 21 CF infants, sample collection at 6 weeks, 4, 6, 9, and 12 months	16S rRNA sequencing (V4–V5 region)	- Reduced bacterial diversity at the end of the first year of life in CF patients - Association of bacterial diversity with CF exacerbation - Reduced abundance of *Bacteroides* and *Roseburia* and increased abundance of *Veillonella* in CF patients	[125]
	13 patients with CF, sample collection from birth to 34 months	16S rRNA sequencing (V4–V6 region)	- Reduced abundance of fecal *Parabacteroides* prior to airway colonization with *Pseudomonas aeruginosa* - Association of intestinal microbiota composition with CF exacerbation	[126]
	27 healthy children, age 8 ± 5 years, 27 CF children, age 8 ± 5 years	16S rRNA sequencing (V1–V3 region)	- Lower bacterial richness and diversity in CF patients - Increased abundance of *Proteobacteria* and *Fusobacteria* in CF patients - Association of intestinal bacteria with fecal calprotectin and lung function	[127]
	25 healthy infants, sample collection at 2, 4, 6, 9 and 12 months 207 CF infants, sample collection at 3, 4, 5, 6, 8, 10, 12 months	16S rRNA sequencing	- Decreased abundance of *Bacteroidetes* and increased abundance of *Proteobacteria* in CF patients - More pronounced intestinal dysbiosis in CF infants with growth retardation	[128]
Respiratory infectious diseases and COVID-19	78 healthy patients, age 46 ± 17 years, 87 COVID-19 patients, age 36 ± 19 years	16S rRNA sequencing	- Reduced abundance of *Actinobacteria* and increased abundance of *Bacteroidetes* in COVID-19 patients - Reduced abundance of *Faecalibacterium prausnitzii* and *Bifidobacterium bifidum* in COVID-19 patients - Association of intestinal microbiota composition with disease severity and systemic immune response (TNF-α, IL-10)	[129]
	15 healthy patients, age 48 (45–48) years, 6 patients with non-COVID pneumonia, age 50 (44–65) years, 15 COVID-19 patients, age 50 (44–68) years	16S rRNA sequencing	- Association of *Firmicutes* with disease severity in COVID-19 patients - Inverse association of *Faecalibacterium prausnitzii* with disease severity in COVID-19 patients - *Bacteroidetes* correlated inversely with fecal SARS-CoV-2 viral load	[130]
	30 healthy controls, age 54 (44–60) years 24 influenza patients, age 49 (33–67) years 30 COVID-19 patients, age 55 (48–62) years	16S rRNA sequencing (V3–V4 region)	- Reduced bacterial diversity in COVID-19 and influenza patients - Intestinal microbiota significantly differs between influenza and COVID-19 patients - Decreased abundance of *Actinobacteria* and *Firmicutes* in influenza patients - Higher relative abundance of opportunistic pathogens such as *Streptococcus*, *Rothia*, *Veillonella,* and *Actinomyces* in COVID-19 patients	[131]
	7 COVID-19 patients with fecal SARS-CoV-2 excretion, age 56 ± 10 years, 8 COVID-19 patients without fecal SARS-CoV-2 excretion, age 50 ± 17 years	16S rRNA sequencing	- Higher abundance of opportunistic pathogens (*Collinsella aerofaciens*, *Collinsella tanakaei*, *Streptococcus infantis*, *Morganella morganii*) in patients with fecal SARS-CoV-2 excretion - Higher abundance of SCFA-producing bacteria (*Bacteroides*, *Parabacteroidetes*, *Lachnospiraceae*) in patients without fecal SARS-CoV-2 excretion	[132]
Asthma	319 children, sample collection at age of 3 and 12 months	16S rRNA sequencing (V3 region)	- No association of bacterial diversity with asthma at age 3 years - Association of low abundance of certain genera (*Faecalibacterium*, *Lachnospira*, *Rothia,* and *Veillonella*,) at 3 months with asthma at age 3 years	[7]
	142 children, age 5 (2–7 years)	16S rRNA sequencing (V4–V6 region)	- Association of usage of macrolides and alteration of intestinal microbiota (increased *Proteobacteria* and *Bacteroidetes*, reduced *Actinobacteria*) with an increased risk of asthma in later life	[133]
	917 children, sample collection at 3 and/or 12 months	16S rRNA sequencing (V4 region)	- Lower bacterial diversity in infants with asthma at age 5 years - Reduced bacterial diversity by antibiotic treatment in early infancy was associated with higher risk of asthma	[134]
	47 children at risk of developing allergy, sample processing at age of 1 week, 1 month and 12 months	16S rDNA (V3/V4 region) 454-pyrosequencing	- Association of low bacterial diversity at age 1 week and 1 month (but not at 12 months) with asthma at age 7 years - No association of asthma in later life with a distinct bacterial phylum	[135]
	27 children with atopic wheeze, 70 healthy control children, age 5 years, sample processing at age of 3 months	16S rRNA sequencing	- No association of atopic wheeze with bacterial diversity - Association of low abundance of *Bifidobacterium* and high abundance of *Streptococcus* and *Bacteroides* with asthma - Association of reduced fecal acetate with asthma	[55]
	298 children, sample collection at age 1 month (n = 130) or 6 months (n = 168)	16S rRNA sequencing	- Association of low abundance of *Bifidobacterium*, *Akkermansia* and *Faecalibacterium* with high risk of atopy and asthma at age 4 years	[136]

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
