# Peer review of "Respiratory and Intestinal Microbiota in Pediatric Lung Diseases—Current Evidence of the Gut–Lung Axis"

_ijms, 2022, doi:10.3390/ijms23126791_

Round 1

Reviewer 1 Report

This is an excellent review paper and I have no serious criticisms regarding interpretations of the cited references which include not only regarding the gut-lung axis, but also the role of respiratory and intestinal microbiota and modulation by the pre-, pro-, and synbiotics in pediatric lung diseases.

However, I add a comment that the role of the intestinal microbiota in Allergic ashma is desirable to reduce the description because the main role of allergy on the disease has widely and firmly believed by the most people.

Author Response

Answer:

First, we would like to thank reviewer 1 for reading the manuscript and making constructive and stimulating comments. As reviewer 1 states correctly, the important role of the intestinal microbiota in the pathophysiology of allergic asthma is widely recognized. However, in our manuscript we aim to critically review and summarize current scientific evidence and to give an overview about the pathophysiological concepts of the host microbiota, even for readers who are not so familiar with this topic. Furthermore, we focus on intestinal dysbiosis in early life, as this factor is thought to exert a negative effect on the development of tolerance to exogenous antigens, thus increasing the risk for asthma and atopy in later life.

Reviewer 1 did not demand any specific changes in the manuscript.

Reviewer 2 Report

The paper by Stricker et al is a review made with the aim to explain the recent mechanistic evidence from animal studies regarding the gut-lung axis and to summarize current knowledge from observational studies and human trials investigating the role of the respiratory and intestinal microbiota and their modulation by pre-, pro-, and synbiotics in pediatric lung diseases.

The paper is well written and the review is well designed.

The paper can be accepted in this form.

Author Response

Answer:

We want to thank reviewer 2 for reading the manuscript and giving feedback. Reviewer view did not demand any changes in the manuscript.
